# Reprimo (RPRM) as a Potential Preventive and Therapeutic Target for Radiation-Induced Brain Injury via Multiple Mechanisms

**DOI:** 10.3390/ijms242317055

**Published:** 2023-12-02

**Authors:** Zhujing Ye, Jin Wang, Wenyu Shi, Zhou Zhou, Yarui Zhang, Jingdong Wang, Hongying Yang

**Affiliations:** State Key Laboratory of Radiation Medicine and Protection, School of Radiation Medicine and Protection, Collaborative Innovation Center of Radiological Medicine of Jiangsu Higher Education Institutions, Suzhou Medical College of Soochow University, Suzhou 215123, China; 18306205857@163.com (Z.Y.); 18862187194@163.com (J.W.); shiwenyv@163.com (W.S.); 18260187268@163.com (Z.Z.); yinxiaolei_69@163.com (Y.Z.); wangjindong@suda.edu.cn (J.W.)

**Keywords:** RPRM knockout mice, radiation-induced brain injury, DNA damage, apoptosis, neuroinflammation, microglial activation, neurons, CCL2, neurogenesis, radiation-induced cognitive impairment

## Abstract

Patients receiving cranial radiotherapy for primary and metastatic brain tumors may experience radiation-induced brain injury (RIBI). Thus far, there has been a lack of effective preventive and therapeutic strategies for RIBI. Due to its complicated underlying pathogenic mechanisms, it is rather difficult to develop a single approach to target them simultaneously. We have recently reported that Reprimo (RPRM), a tumor suppressor gene, is a critical player in DNA damage repair, and RPRM deletion significantly confers radioresistance to mice. Herein, by using an RPRM knockout (KO) mouse model established in our laboratory, we found that RPRM deletion alleviated RIBI in mice via targeting its multiple underlying mechanisms. Specifically, RPRM knockout significantly reduced hippocampal DNA damage and apoptosis shortly after mice were exposed to whole-brain irradiation (WBI). For the late-delayed effect of WBI, RPRM knockout obviously ameliorated a radiation-induced decline in neurocognitive function and dramatically diminished WBI-induced neurogenesis inhibition. Moreover, RPRM KO mice exhibited a significantly lower level of acute and chronic inflammation response and microglial activation than wild-type (WT) mice post-WBI. Finally, we uncovered that RPRM knockout not only protected microglia against radiation-induced damage, thus preventing microglial activation, but also protected neurons and decreased the induction of CCL2 in neurons after irradiation, in turn attenuating the activation of microglial cells nearby through paracrine CCL2. Taken together, our results indicate that RPRM plays a crucial role in the occurrence of RIBI, suggesting that RPRM may serve as a novel potential target for the prevention and treatment of RIBI.

## 1. Introduction

Radiation-induced brain injury (RIBI), an adverse effect of cranial radiotherapy (CRT) for patients with primary and metastatic brain tumors as well as head and neck cancer, results from the combination of DNA damage, oxidative stress, cell death, inflammatory response, etc., in the brain after exposure to irradiation, leading to acute, early-delayed and late-delayed symptoms. While acute and early-delayed RIBI are usually transient and reversible, late-delayed RIBI, e.g., radiation-induced neurocognitive impairment, is progressive and irreversible [1]. With the significantly prolonged survival in cancer patients after receiving CRT with advanced techniques, late-delayed RIBI has been increasingly recognized. Up to 50–90% of patients who survive longer than 6 months after whole-brain radiation therapy (WBRT) develop cognitive impairment [2,3], which seriously deteriorates their quality of life. The mechanisms underlying radiation-induced cognitive impairment include hippocampal neurogenesis inhibition, neuroinflammation, altered neuronal function and vascular and glial cell clonogenic populations after exposure to ionizing radiation (IR) [4,5]. It has been demonstrated that attenuating radiation-induced neurogenesis inhibition through hippocampus avoidance during WBRT, the administration of drugs such as Baicalein and lithium, the transplantation of neural stem cells (NSCs) or their secreted exosomes ameliorates cognitive decline [6,7,8,9,10,11,12]. Targeting the neuroinflammatory microenvironment using antioxidants, renin–angiotensin system (RAS) blockers, etc., has also been shown to prevent/ameliorate radiation-induced cognitive impairment in animal models [13,14,15,16]. Despite the progress made so far, there is still a lack of effective preventive and therapeutic strategy for RIBI; thus, more investigation is required.

Reprimo (RPRM) is a tumor-suppressor gene that has been found to be involved in the progression of a variety of malignant tumors such as pituitary tumors, gastric cancer, etc., and thus may serve as a biomarker for early cancer detection [17,18,19,20]. RPRM is variably expressed in different tissues, in which the brain is one of the organs with the most abundant RPRM expression [20,21]. Although it has been suggested that RPRM may be essential for the development and function of the brain [21], it remains unknown whether RPRM is related to any pathological brain phenotypes. Interestingly, RPRM can be induced by DNA damage [17,19]. We have recently discovered that RPRM indeed plays an important role in the DNA damage repair and radiosensitivity of cells through its negative regulatory effect on the ataxia–telangiectasia-mutated (ATM) protein kinase [22]. Thus, RPRM knockout significantly delays death in mice exposed to whole-body X-irradiation, and alleviates radiation-induced intestinal and hematopoietic system injury [22,23], suggesting that RPRM may play a crucial role in radiation-induced tissue injury. However, whether RPRM is involved in the occurrence of RIBI remains unknown.

To further elucidate the role of RPRM in radiation-induced tissue injury, here in this study, we investigated how RPRM deletion affected RIBI using our established RPRM knockout mouse model. We demonstrated that RPRM knockout not only significantly mitigated hippocampal DNA damage and apoptosis shortly after mice were exposed to whole-brain irradiation (WBI), but also attenuated the WBI-induced inhibition of neurogenesis and decline in cognition in the long term. Furthermore, we found that RPRM knockout dramatically mitigated neuroinflammatory response and microglial activation induced by WBI, and the attenuated microglial activation by RPRM knockout was associated with its protective effect on both microglia and neurons.

## 2. Results

### 2.1. RPRM Knockout Mitigates Radiation-Induced Acute Hippocampal DNA Damage and Apoptosis

We have previously revealed a vital role of RPRM in DNA damage repair [22]. To determine whether this role of RPRM impacts RIBI, we first compared the level of hippocampal DNA damage in WT and KO mice shortly after WBI using γ-H2AX as the marker for DNA damage [24]. As expected, 10 Gy WBI induced a dramatic elevation of γ-H2AX levels in the hippocampi of both WT and RPRM KO mice shortly after irradiation. But the γ-H2AX level was much lower in RPRM KO mice than in WT mice (Figure 1A,B), suggesting a lower level of DNA damage in irradiated RPRM KO mice.

Cytotoxicity is the primary reason for radiation-induced acute tissue injury, and it has been found that RPRM overexpression enhances DNA damage-induced apoptosis, which RPRM knockdown inhibits [19]. To determine whether RPRM knockout reduces WBI-induced cytotoxicity in the brain, we detected hippocampal apoptosis 6 h after WBI using immunohistochemical staining for cleaved caspase-3. As shown in Figure 1C,D, WBI induced a great increase in the number of cells positive for cleaved caspase-3 in the hippocampi of both WT and RPRM KO mice, but there were much less cleaved caspase-3 positive cells in irradiated RPRM KO mice than in irradiated WT mice, suggesting a lower level of apoptosis in irradiated RPRM KO mice.

All these data demonstrate that RPRM knockout obviously decreased radiation-induced DNA damage and apoptosis in the mouse brain, indicating that RPRM knockout mitigates radiation-induced acute brain injury.

### 2.2. RPRM Knockout Ameliorates Radiation-Induced Cognitive Impairment

To confirm the potential role of RPRM in RIBI, we further investigated how RPRM deletion would affect late-delayed RIBI, i.e., radiation-induced cognitive impairment. We found that RPRM knockout did not significantly affect the locomotor activity and anxiety level in mice before and after WBI (Appendix A). MWM results showed that the swimming speed of mice was not much different with and without RPRM 50–54 days after WBI (Figure 2A). However, on the fifth day in the place navigation test, in contrast with the irradiated WT mice that needed greater latency than sham-irradiated WT mice, i.e., the time mice spent finding the hidden platform after 4 days of training (Figure 2B), irradiated RPRM KO mice did not show an obvious increase in the latency compared with their sham-irradiated controls (Figure 2C). Moreover, in the spatial probe test, while irradiated WT male mice spent significantly less time in the target quadrant and crossed the target quadrant significantly less frequently than sham-irradiated WT mice, the reduction in the time RPRM KO mice spent in the target quadrant and their frequency of crossing the target quadrant after irradiation were much smaller and less significant (Figure 2D,E and Appendix A), suggesting that WBI caused a less severe cognition decline in RPRM KO mice than in WT mice. Although we also noticed that RPRM deletion alone appeared to have a tendency to decrease the cognition of mice (Figure 2B–E), these data indicate that RPRM knockout ameliorates the loss of learning and memory ability of mice caused by irradiation.

### 2.3. RPRM Knockout Attenuates Radiation-Induced Neurogenesis Inhibition

Neurogenesis inhibition caused by IR is one of the primary mechanisms underlying radiation-induced cognitive impairment [4]. Thus, we compared the neurogenesis inhibition in RPRM KO and WT mice after WBI. RPRM KO mice showed a slight and statistically insignificant reduction in neurogenesis when compared with WT mice without IR (Figure 3A–C), which was in accordance with the decreasing tendency in the cognition of unirradiated RPRM KO mice (Figure 2B–E). However, at 2 months post-WBI, compared with sham-irradiated controls, the numbers of BrdU+ cells and BrdU+/NeuN+ mature neurons decreased by 50% and 59%, respectively, in irradiated WT mice, but decreased only by 17% and 19%, respectively, in irradiated RPRM KO mice (Figure 3A–C). This indicated that RPRM deletion significantly attenuated radiation-induced neurogenesis inhibition in mice. Thus, RPRM KO mice had more newborn cells and neurons than WT mice after exposure to WBI (Figure 3A–C). Taken together, all these data indicate that RPRM knockout ameliorates long-term RIBI.

### 2.4. RPRM Knockout Diminishes Radiation-Induced Neuroinflammation

DNA damage and oxidative stress induced by IR lead to a pro-inflammatory environment in the brain that contributes to radiation-induced cognitive impairment [3,4]. Thus, we further examined whether RPRM deletion reduced radiation-induced inflammatory response. As shown in Figure 4A–E, WBI caused a great increase in the expression of pro-inflammatory cytokines, including IL-1α, IL-1β, TNF-α, CXCL10 and CCL2, in the hippocampi of WT mice shortly after IR; however, RPRM deletion dramatically attenuated the increase. In particular, there was no statistically significant increase in the level of CCL2 in RPRM KO mice after WBI.

IR-induced inflammatory response in the brain is also mediated by the activation of microglia, the resident immune cells of the central nervous system (CNS), which respond to IR-induced microglial damage as well as neuronal damage. As shown in Figure 4F,G, six hours after WBI, there was a dramatic increase in the number of amoeboid microglia (Appendix A) in the hippocampi of WT mice, but the increase was much lower when RPRM was deficient, suggesting that microglia were significantly activated by irradiation in WT mice, but the activation was inhibited by RPRM knockout.

Moreover, two months after radiation exposure, compared with their relative sham-irradiated control, irradiated WT mice still had a significantly greater number of amoeboid microglia in their hippocampi and displayed a significant and persistent elevation in the levels of TNF-α and CCL2, while irradiated RPRM KO mice only showed an upward trend but without statistical significance in the number of amoeboid microglia and in the expression of TNF-α and CCL2 (Figure 5). The data suggested that RPRM knockout attenuated WBI-induced chronic microglial activation and inflammatory response. Taken together, all these results indicate that RPRM knockout diminishes radiation-induced acute and chronic neuroinflammation.

### 2.5. RPRM Knockout Protects Microglia against Radiation-Induced Damage and Attenuates Their Activation Both In Vivo and In Vitro

Due to the critical role of microglial activation in RIBI [25], we further explored how RPRM knockout diminished WBI-induced microglial activation. We first demonstrated that RPRM deletion protected microglia against radiation-induced DNA damage both in vivo and in vitro. Six hours after WBI, although there was a great increase in the number of Iba-1+/γH2AX+ cells in the hippocampi of both WT and RPRM KO mice, RPRM KO mice had only approximately half the number of Iba-1+/γH2AX+ cells as WT mice (Figure 6A,B). Similarly, as shown in Figure 6C,D, the fluorescence intensity of γH2AX significantly increased in WT primary microglial cells in vitro 6 h after irradiation, but RPRM knockout reduced the γH2AX intensity by 44%, indicating a protective effect of RPRM in irradiated microglia.

We next confirmed that RPRM deletion significantly reduced microglial activation induced by irradiation both in vivo and in vitro by using dual immunofluorescence staining for both Iba-1 and CD68, a marker for microglial activation [26]. Six hours after WBI, the number of Iba-1+/CD68+ cells increased dramatically in the hippocampi of WT mice, but only increased insignificantly in RPRM KO mice, and WT mice had 3.5 times as many Iba-1+/CD68+ cells as RPRM KO mice (Figure 6E,F). This is in agreement with the microglial morphological results from Figure 4F,G. Primary microglia cultured in vitro were also activated by irradiation. And the CD68 level of RPRM^+/+^ microglia was 86% higher than that of RPRM^−/−^ microglia after radiation exposure (Figure 6G,H), indicating that RPRM deletion attenuated the activation of primary microglia induced by irradiation.

Interestingly, we also found that RPRM deletion reduced the expression level and secreted amount of CCL2 of primary microglia after irradiation. As shown in Figure 6I–K, IR caused a significant induction of CCL2 expression in microglia and enhanced its secretion, but RPRM knockout inhibited both. Compared with RPRM^+/+^ microglia after irradiation, irradiated RPRM^−/−^ microglia exhibited a 62% lower CCL2 expression and secreted 32% less CCL2. The data agreed with what we observed in the mouse hippocampus (Figure 4E). All these results confirm that RPRM deletion has a protective effect on microglia and an inhibitory effect on microglial activation after exposure to IR, thus mitigating IR-induced neuroinflammation.

### 2.6. RPRM Knockout Protects Neurons against Radiation-Induced Damage Both In Vivo and In Vitro

As fundamental cells of the brain, neurons are also subjected to IR-induced damage. Due to the critical role of RPRM in DNA damage repair [22] and DNA damage-induced apoptosis [19], it was not unexpected that less severe neuronal DNA damage and apoptosis were observed in the hippocampi of RPRM KO mice than in WT mice after exposure to 10 Gy WBI (Figure 7A–D). In detail, compared with WT mice, the γH2AX fluorescence intensity of NeuN+ neuronal cells and the number of NeuN+ neuronal cells which was positive for cleaved caspase-3 in the hippocampi of RPRM KO mice were, respectively, reduced by 19% and 62% at 6 h post-WBI (Figure 7A–D). The attenuation of radiation-induced DNA damage and apoptosis via RPRM deletion was also confirmed in primary neurons cultured in vitro (Figure 7E–H). After exposure to X-irradiation at a dose as high as 50 Gy, RPRM deletion reduced the DNA damage by 25% 6 h post-IR (Figure 7E,F). Primary neurons cultured in vitro were invulnerable to radiation-induced apoptosis, and RPRM deletion further enhanced their radioresistance. Six hours after exposure to 50 Gy X-irradiation, the level of apoptosis was 24% lower in RPRM^−/−^ neurons than in RPRM^+/+^ neurons (Figure 7G,H). Moreover, we found that RPRM knockout helped preserve the structure of neurons. Three days after exposure to 10 Gy X-irradiation, the number of neuronal dendrites was significantly reduced by 34% for RPRM^+/+^ neurons, but only insignificantly reduced by 18% for RPRM^−/−^ neurons (Figure 7I,J). Ten days after irradiation, in contrast to the severely broken dendrites of RPRM^+/+^ neurons, RPRM^−/−^ neurons still remained relatively intact and retained their complex structure (Figure 7K). All these data indicate that RPRM knockout attenuates radiation-induced neuronal damage and help preserve neuronal structure.

### 2.7. RPRM Knockout Diminishes the Activation of BV2 Cells Caused by Irradiated Primary Neurons, Which Involves Paracrine CCL2

In addition to being activated by their own DNA damage, microglia are also activated in response to neuronal damage caused by IR. Since RPRM knockout protected neurons against radiation-induced damage, this may lead to a mitigation in the microglial activation induced by damaged neurons. In fact, we found that the activation of BV2 cells, an immortalized murine microglial cell line, was reduced by 24% and 40% after co-culture with irradiated primary RPRM^−/−^ neurons for 1 and 6 h, respectively, when compared with the BV2 cells co-cultured with irradiated RPRM^+/+^ neurons (Figure 8A–C).

We further explored what was responsible for the reduction in the activation of BV2 cells after co-culture with irradiated RPRM^−/−^ neurons. It has been reported that the CCL2 produced by neurons was a key mediator of microglial activation [27], and microglial activation/migration was attenuated in MCP1^−/−^ mice [28]. Furthermore, we have already found that RPRM knockout almost abolished the elevation of the expression of CCL2 in the mouse hippocampus after exposure to WBI (Figure 4E). RPRM deletion also dramatically reduced the expression level and secreted amount of CCL2 in primary microglia after irradiation (Figure 6I–K). Thus, we examined whether RPRM deletion reduced the CCL2 expression and secretion in irradiated primary neurons. As shown in Figure 8D–G, CCL2 was induced in neurons through irradiation, and RPRM deletion, respectively, reduced the CCL2 level of neurons by 37% and 64% at the mRNA and protein level 6 h after irradiation. Furthermore, in contrast to an obvious elevation in CCL2 secretion by primary RPRM^+/+^ neurons after irradiation, irradiated primary RPRM^−/−^ neurons showed little increase in CCL2 secretion compared with sham-irradiated control. To further prove that paracrine CCL2 may be involved in the reduction in microglial activation after co-culture with irradiated neurons by RPRM knockout, on one hand, we confirmed that CCL2 recombinant protein itself could activate BV2 cells manifested by a significant increase in the level of CD86, a widely used marker for the M1 phenotype of microglia [29] (Figure 8H,I); on the other hand, when CCL2 antibody was added into the co-culture of irradiated primary RPRM^+/+^ neurons and BV2 cells, the activation of BV2 was attenuated by 34% (Figure 8J,K). All these data indicate that the reduction in microglial activation via RPRM deletion is partially ascribed to its impact on neurons, i.e., RPRM deletion reduces the induction of CCL2 expression in neurons, and its secretion after IR in turn mitigates the activation of nearby microglia in a paracrine manner.

## 3. Discussion

The occurrence of RIBI involves DNA damage, cytotoxicity and cell loss shortly after irradiation, as well as complicated dynamic interaction between different cell types such as microglia, neurons, etc., which lead to the loss of hippocampal neurogenesis, microglial activation, neuroinflammation and the alteration of neuronal structure and function, which in turn result in a long-term effect, i.e., cognitive impairment [1,4,5]. Therefore, from the point view of developing countermeasures for RIBI, it would be ideal to combine strategies that can promote DNA damage repair, reduce cytotoxicity and cell loss and mitigate neuroinflammation [30]. Intriguingly, here in this study, we demonstrated that the deletion of RPRM may achieve these goals simultaneously.

We have recently reported that, while the overexpression of RPRM attenuates DNA damage repair through inhibiting both homologous recombination (HR) and nonhomologous end-joining (NHEJ) pathways as well as increases cellular radiosensitivity, a deficiency in RPRM mitigates IR-induced DNA damage accumulation and promotes cell survival [22]. Thus, it was not surprising when, here, we found that RPRM deletion significantly reduced WBI-induced DNA damage and apoptosis in the hippocampus (Figure 1) due to its abundant expression in the brain [20,21]. We further demonstrated that RPRM deletion protected microglia and neurons against IR-induced DNA damage and apoptosis both in vivo and in vitro (Figure 6 and Figure 7). These results confirm the universality of the function of RPRM in DNA damage repair regardless of cell type, including post-mitotic cells like neurons. Our results also confirm the role of RPRM in DNA damage-induced apoptosis [19]. Additionally, cranial irradiation causes similar types of neuronal architectural changes found in many neurodegenerative diseases, contributing to the impairment of learning and memory [31]. Here, we found that RPRM deletion could diminish the destruction of neuronal dendrites and preserve neuronal architecture better after IR (Figure 7I–K). Moreover, given the protective effect of RPRM deletion on both microglia and neurons, one could expect that RPRM knockout may also protect NSCs against IR-induced DNA damage and apoptosis. We indeed observed a dramatically attenuated neurogenesis inhibition in RPRM KO mice after exposure to WBI (Figure 3). This is in accordance with what we found in the hematopoietic system, in which RPRM deletion preserved hematopoietic regeneration through promoting the DNA repair and proliferation of hematopoietic stem cell post IR [23]. All these data suggest that RPRM deletion may protect different types of brain cells against irradiation-induced damage.

We also demonstrated that RPRM deletion significantly attenuated IR-induced microglial activation (Figure 6). As a key mediator of neuroinflammation, microglia are activated quickly upon IR exposure, transiting to the M1 pro-inflammatory phenotype [26,32]. Their activation by IR can be through both direct and indirect ways. IR-induced DNA damage directly activates microglia via an activation of transcription factors such as nuclear factor κB, cAMP response element-binding protein and activating protein 1, resulting in an elevation of the expression of inflammatory factors, including IL-1β, TNF-α and MCP-1/CCL2 [33,34]. Thus, microglial activation was attenuated when RPRM deletion reduced irradiation-induced damage to microglia. Microglia can also be activated by IR-damaged neurons through released “danger” signal molecules such as ATP [35,36]. Here, we confirmed that CCL2, a CC chemokine that can be induced by irradiation [37,38], was one kind of the “danger” signal molecules. Therefore, when RPRM deletion reduced the amount of CCL2 secreted by irradiated neurons and microglia, it led to a weakened microglial activation (Figure 6 and Figure 8). Even 2 months after irradiation, unlike WT mice, RPRM knockout mice did not show significant microglial activation (Figure 5). These data indicate that RPRM deletion attenuates microglial activation through its protective effect on both microglia and neurons. This in turn also facilitates the mitigation of radiation-induced neurogenesis inhibition [25].

Moreover, since RPRM deletion protected neurons and microglia against irradiation-induced damage and activation, it would reduce the secretion of pro-inflammatory cytokines by the damaged neurons and activated microglia [39], which is supported by the significant reduction in the expression of pro-inflammatory cytokines, including IL-1α, IL-1β, TNF-α, CXCL10 and CCL2, in the hippocampi of irradiated RPRM KO mice (Figure 4A–E). In addition to its role in microglial activation [27], CCL2 participates in IR-induced immune response such as promoting macrophage infiltration, recruiting myeloid-derived suppressor cells and T cells through binding to its receptors, including CCR1, CCR2, CCR4 and CCR5 [40,41]. CCL2 has been found to play an important role in radiation-induced brain, lung and liver injury [41]. Interestingly, in a cranial irradiated mouse model, CCL2 did not recruit monocytes to injury sites; instead, it affected neuroinflammation and neurogenesis. The loss of CCL2 restored hippocampal neurogenesis after irradiation [42]. Moreover, CCR2 deletion prevented neuronal and hippocampus-dependent cognition dysfunction induced by cranial irradiation [43]. Therefore, CCL2 has been proposed to be a marker for radiation-induced injury and a therapeutic target [41]. Here, we demonstrated that RPRM deficiency significantly decreased the CCL2 induction after radiation exposure in both primary microglia and neurons in vitro (Figure 6I–K and Figure 8D–G) and the mouse brain in vivo (Figure 4E), suggesting that RPRM plays a critical role in RIBI. RPRM deletion dramatically mitigates radiation-induced neuroinflammation, which in turn contributes to the attenuation of radiation-induced neurogenesis inhibition and cognitive impairment in RPRM KO mice. These results suggest that RPRM is involved in inflammation. However, although a recent study identified RPRM as an immune-related gene [44], we did not observe any difference in the expression of those pro-inflammatory cytokines and the level of microglial activation between WT and RPRM KO mice without irradiation (Figure 4 and Figure 5). Given the pivotal role of RPRM in DNA damage repair [22], the modulation of radiation-induced neuroinflammation by RPRM may be related to the activation of nucleic acid-sensing pathways [45], which is worthy of further investigation.

As a result, RPRM knockout obviously attenuated the decline in the cognitive ability of mice after irradiation (Figure 2). However, it was noted that RPRM deletion alone showed a tendency to cause cognitive decline in mice (Figure 2). Additionally, the tendency of cognitive decline in RPRM KO mice coincided with the decreasing trend in their neurogenesis (Figure 3). This may be related to a potential role of RPRM during the self-renewal of NSCs [46]. All these results suggest that RPRM, a poorly understood protein, may play important roles not only in the development and function of CNS, but also in RIBI. Thus, further studies are required.

Due to its critical role in RIBI, it is worth further exploring the potential implications of RPRM. For example, can the level of RPRM in the brain (ideally the RPRM in serum, plasma or cerebrospinal fluid in which RPRM can be easily measured) be used as an index predicting the risk of RIBI? Are there any compounds or methods that can effectively decrease the level of RPRM in the normal tissues so that they can be used as radioprotectors? Moreover, we also need to clarify the toxicity of RPRM deletion. Would transient RPRM deletion cause any immediate toxicity or side-effects to humans? Since RPRM is a tumor suppressor [18,19,20], would transient RPRM deletion increase the risk of cancer incidence, as well as health issues other than cancer in the long run? All these questions need further investigation.

Finally, RPRM is an estrogen-repressed gene [47,48]. RPRM has also been found to play a role in temperature regulation in a sex-dependent manner, but its expression was female-biased in the neurons of ventromedial hypothalami of mice, which was repressed by estrogen signaling during male development [49]. These studies indicate a sex difference of RPRM expression and function. Therefore, although female mice were reported to be more resistant to the adverse neurocognitive effects of radiation than male mice [50], in order to develop an RPRM-based countermeasure for RIBI, whether RPRM knockout plays a similar attenuating effect on RIBI in female mice needs to be clarified in the near future.

## 4. Methods and Materials

### 4.1. Animal Model and Irradiation

All animal experiments were approved by the ethics committee of Soochow University. The Medical Experimental Animal Care Guidelines of Soochow University based on the National Animal Ethical Polices of China was strictly complied with in all animal experiments. Male RPRM gene knockout C57BL/6J mice (KO) constructed in our laboratory and their counterpart wild-type mice (WT) at the age of 6–8 weeks were used. All mice were kept in the specific pathogen-free animal facility of Soochow University Experimental Animal Center. Mice were randomly divided into different groups, including sham-irradiated control groups and irradiated groups, at different times post-irradiation, where each group contained 3–12 mice. Mice in irradiated groups were whole-brain irradiated with a single dose of 10 Gy X-rays (160 kVp, RAD SOURCE RS2000 X-ray machine, Rad Source Technologies Inc., Buford, GA, USA) at a dose rate of 1.16 Gy/min at room temperature (RT).

### 4.2. Cell Culture

Primary neurons were isolated from the fetal cerebral cortex of C57BL/6 mice on the 14th day of pregnancy. The cerebral cortex dissected from embryos was digested, and the obtained cell pellets were resuspended in NEUROBASAL MED SFM medium (Thermo-Fisher Scientific, Waltham, MA, USA) supplemented with 2% B27 (Gibco, Grand Island, NY, USA), 1 mM L-glutamine (Beyotime Biotech. Inc., Shanghai, China) and 0.5% penicillin–streptomycin (Beyotime, Shanghai, China). Then, the cell suspension was filtered through a 40 μm pore-size cell filter. The obtained cells were seeded in a 6-well plate at a density of 2 × 10^5^ cells/mL and cultured in an incubator. One-half of the medium was removed and replaced with an equal volume of fresh medium every 3 days. The primary neurons were used for neuronal dendrite detection and other experiments after 6 and 11 days of culture, respectively. Primary microglia were isolated from the cerebral cortex of newborn mice within 24 h after birth. As described above, the cerebral cortex from newborn mice was digested, and the obtained cell pellets were resuspended in DMEM-F12 proliferation medium (HyClone, South Logan, UT, USA) supplemented with 0.5% FBS and 1% penicillin–streptomycin. Then, the cell suspension was filtered though a 40 μm pore-size cell filter. The obtained cells were seeded in a T25 culture flask at a density of 2 × 10^5^ cells/mL and cultured in an incubator. One-half of the DMEM-F12 medium was changed every 3 days. After 14 days of culture, the mixed glia were trypsinized with 0.05% trypsin (Beyotime, Shanghai, China) for 10 min at 37 °C, and primary microglia were separated by gently tapping the side wall of the culture flask 10 times. The isolated primary microglia were reseeded in a 12-well plate at a density of 1 × 10^5^ cells/mL and cultured in an incubator. One-half of the DMEM-F12 medium was changed every 3 days, and the primary microglia were used for experiments after 12 days of culture. BV2 microglial cells purchased from Procell Life Science & Technology Co. Ltd. (Wuhan, China) were cultured in MEM medium (Procell, Wuhan, China) containing FBS (10%, Gibco, Carlsbad, CA, USA) and penicillin–streptomycin (1%, Beyotime, Shanghai, China) in an incubator with 5% CO_2_ at 37 °C.

### 4.3. Cell Irradiation and Co-Culture System

Primary microglial cells of 1 × 10^5^ were plated on coverslips in a 6-well plate. Twenty-four hours later, microglial cells were irradiated with 10 Gy of X-rays at 1.16 Gy/min at RT. On the 7th and 12th day after 2 × 10^5^ primary neuronal cells were plated, the neurons were irradiated with 10 and 50 Gy of X-rays depending on the purpose of experiments. For co-culture experiment, 2 × 10^5^ BV2 microglial cells were plated on coverslips in a 6-well plate 24 h before co-culture; immediately after irradiation, the primary neuronal cells which were X-irradiated with 50 Gy were put into co-culture with unirradiated BV2 microglial cells in NEUROBASAL MED SFM medium using a transwell co-culture membrane insert with a pore size of 0.4 μm (Millipore, Billerica, MA, USA) until analysis.

### 4.4. Western Blotting

Mice were sacrificed 1, 6 and 24 h after 10 Gy WBI. The hippocampal tissue was isolated and lysed via sonication in RIPA lysis buffer containing 1% protease inhibitor cocktail for 60 s at 4 °C. The homogenates were centrifuged at 16,000× *g* for 20 min at 4 °C. Proteins were separated on a 12% SDS-PAGE gel, then transferred to a polyvinylidene difluoride membrane (Bio-Rad, Hercules, CA, USA). After blocking with TBST containing 5% skim milk for 2 h at RT, the membranes were, respectively, incubated with rabbit anti-γH2AX (1: 1000; Abcam, Cambridge, MA, USA) and mouse anti-β-actin (1:500; Beyotime, Shanghai, China) antibodies overnight at 4 °C, followed by incubation with the corresponding HRP-conjugated secondary antibodies (1:500; Beyotime, Shanghai, China) for 2 h at RT. Proteins of interest were detected on a FluorChem™ M System (ProteinSimple, Santa Clara, CA, USA) after the membranes were processed with ECL-plus (Beyotime, Shanghai, China). Images were quantified using Image J (NIH, Bethesda, MD, USA).

### 4.5. Immunohistochemistry

At different times after 10 Gy WBI, mice were anesthetized and transcardially perfused with PBS, followed by 4% paraformaldehyde (Beyotime, Shanghai, China). The extracted brain was fixed with 4% paraformaldehyde overnight and sequentially dehydrated in 15% and 30% sucrose solution at 4 °C for 1–2 days. The tissue samples were then embedded in OCT tissue-embedding agent (Sakura, Torrance, CA, USA) and quickly frozen at −80 °C. The frozen tissues were cut into 20 μm thick sections using a cryostat (Leica, Weztlar, Germany). Every tenth coronal frozen brain section was selected. After being treated with 3% H_2_O_2_ for 20 min at 95 °C to block endogenous peroxidase activity, the section was blocked with 10% goat serum albumin (Boster, Wuhan, China) in PBS for 60 min at RT, then incubated overnight at 4 °C with rabbit anti-cleaved caspase-3 (1:500; Cell Signaling Technology, Danvers, MA, USA) and rabbit anti-Iba-1 (1:500; Cell Signaling Technology, Danvers, MA, USA) antibodies, respectively, followed by incubation with corresponding secondary antibodies (1:500; Beyotime, Shanghai, China) for 1 h at RT. After being stained in DAB (ZSGB-BIO, Beijing, China) for 3–6 min, the section was washed with clean water and stained in hematoxylin (Cancercell, Wuhan, China) for 25 s. Following sealing, the section was observed under a microscope (Olympus, IX73, Tokyo, Japan), and the number of brown positive cells located within the granular cell layer (GCL) and subgranular zone (SGZ) in the dentate gyrus (DG) of the hippocampus was counted under bright field. Five sections were examined and quantified for each mouse brain. The results are expressed as the total number of positive cells of five sections.

### 4.6. BrdU Incorporation

RPRM KO mice and WT mice were treated with 10 Gy WBI. Three weeks later, they were intraperitoneally injected with BrdU (Sigma-Aldrich, St. Louis, MO, USA) at 50 mg/kg every day for 7 consecutive days. Brain tissue samples were processed 4 weeks after BrdU incorporation for neurogenesis analysis using immunofluorescence microscopy.

### 4.7. Immunofluorescence Microscopy

For brain tissues, after blocking with 5% goat serum in PBS containing 0.3% Triton X-100 for 60 min at RT, brain sections were incubated overnight at 4 °C with rabbit anti-BrdU, mouse anti-NeuN, rabbit anti-cleaved caspase-3, rabbit anti-Iba-1 and rabbit anti-CD68 antibodies (all from Cell Signaling Technology, Danvers, MA, USA, 1:500), respectively, followed by incubation for 60 min at RT with Alexa Fluor 488 and Cy3-labeled goat anti-mouse and goat anti-rabbit IgG(H + L) (1:500; Beyotime, Shanghai, China), respectively. The sections were then counterstained with DAPI (5 µg/mL) for 5 min at RT. The samples were observed under a confocal scanning laser fluorescence microscope (Olympus, FV1200, Tokyo, Japan). The total number of cells with positive staining in the SGZ and GCL of the DG was obtained by combining the counts from all five analyzed sections of each mouse brain.

For cultured cells, cells on slides were fixed in 4% paraformaldehyde for 15 min at RT. After permeabilization with 5% Triton X-100 at 4 °C for 15 min and blocking with 5% goat serum in PBS containing 0.1% Triton X-100 for 60 min at RT, cells were, respectively, incubated overnight at 4 °C with the following primary antibodies: rabbit anti-Iba-1, rabbit anti-CD86, rabbit anti-cleaved caspase-3 antibodies (all three from Cell Signaling Technology, Danvers, MA, USA, 1:500), rabbit anti-CCL2 antibody (Beyotime, Shanghai, China, 1:500), rabbit anti-beta Tubulin 3/Tuj1 and mouse anti-γH2AX antibody (Both from GeneTex, Irvine, CA, USA; 1:500). Then, the cells were incubated for 60 min at RT with Alexa Fluor 488 and Cy3-labeled goat anti-mouse and goat anti-rabbit IgG(H + L) (Beyotime, Shanghai, China,1:500), respectively. Following DAPI (5 μg/mL) counterstaining for 5 min, cells were observed under a fluorescence microscope (FV1200, Olympus, Tokyo, Japan). At least 300 primary microglia, 100 primary neurons and 500 BV2 cells per sample were examined.

### 4.8. Enzyme-Linked Immunosorbent Assay (ELISA)

Six hours after irradiation, the culture medium for primary microglial and neuronal cells was collected and examined using mouse monocyte chemotactic protein 1/monocyte chemotactic and activating factor (MCP-1/MCAF) ELISA kit (Yipu, Wuhan, China) according to manufacturer’s instructions. Briefly, 10 μL of culture supernatant was mixed with sample diluent, enzyme-labeled reagent and chromogenic agent, successively. Then, absorbance at 450 nm was measured using a Multimode Plate Reader (Synergy 2, BIO-TEK, Winooski, VT, USA).

### 4.9. Quantitative Real-Time Polymerase Chain Reaction (qRT-PCR)

Total RNA was extracted from the hippocampal tissue using Tissue RNA Purification Kit Plus (YISAHAN, Shanghai, China). The cDNA was synthesized from total RNA by using 5× All-In-One RT MasterMix (ABM, Richmond, Canada) following the standard protocols. Quantitative PCR was performed by using the synthetic primers and SYBR Green PCR Master Mix (Monad, Suzhou, China). The sequences of primers used are listed in Table 1. GAPDH was used as a housekeeping gene to determine the relative mRNA levels of these pro-inflammatory genes in WT and KO mice.

### 4.10. Open Field Test (OF) and Morris Water Maze (MWM)

Open field test was carried out 7 weeks after 10 Gy WBI. RPRM KO mice and WT mice each were placed in the center of the open field apparatus (Jiliang, DigBehv-MG, Shanghai, China). An 80 cm × 80 cm square, 10 cm away from the walls, was defined as the central region. The movement of mice was recorded for 15 min using a video-imaging system (Jiliang, DigBehv-MG, Shanghai, China). The distance the mice traveled, their mean speed and the time they spent in the central region were analyzed.

Following open field test, the MWM was carried out as described previously [51]. The place navigation test was performed in 5 consecutive days, in which mice were trained to use the cues available in the testing room to locate the submerged hidden platform. The mean speed of the mice and the time the mice spent searching the platform were recorded and analyzed using Jiliang Software JL2.0 ((Jiliang, DigBehv-MG, Shanghai, China). On the following day, after place navigation test, a 60 s spatial probe test was conducted. After removing the submerged platform, mice were placed into the tank from the opposite of the quadrant where the submerged platform was, and then the time the mice spent crossing the target quadrant and all four quadrants along with the frequency of mice entering the target quadrant were recorded and analyzed.

### 4.11. Statistical Analysis

All statistical analyses were performed using Origin 2019 software. Data are expressed as the means of at least three independent experiments or mouse samples ± standard error (SEM). Statistical significance was calculated using two-way ANOVA with Tukey’s correction or paired *t*-test. A *p* value of <0.05 indicated a statistically significant difference between groups.

## 5. Conclusions

In summary, using our established RPRM knockout mouse model, we clearly demonstrated that RPRM deletion significantly mitigates acute and long-term RIBI. Specifically, RPRM deficiency not only decreases IR-induced hippocampal DNA damage and apoptosis, attenuates microglial activation and neuroinflammation, but also relieves neurogenesis loss and protects neurons against IR-induced damage, thus attenuating the decline in cognition induced by cranial irradiation. These results shed light on the crucial role of RPRM in the occurrence of RIBI, suggesting its potential of serving as a therapeutic target for RIBI.

## Figures and Tables

**Figure 1 ijms-24-17055-f001:**
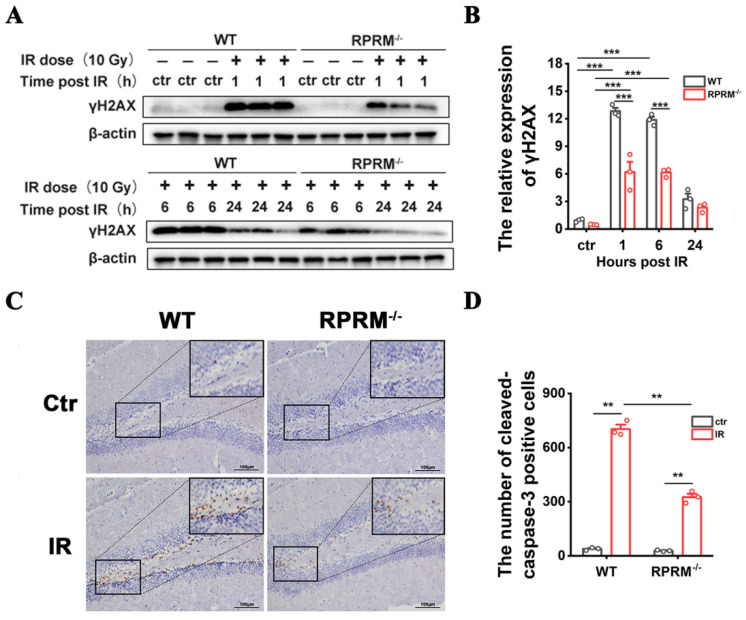
RPRM knockout mitigates WBI-induced acute hippocampal DNA damage and apoptosis. (**A**,**B**) Western blotting images of the γH2AX expression in the hippocampi of WT and RPRM KO mice at different times after exposure to 10 Gy WBI and quantification of γH2AX level. *n* = 3. (**C**,**D**) Immunohistochemistry images of cleaved caspase-3 in the hippocampi of WT and RPRM KO mice at 6 h after exposure to 10 Gy WBI and quantification of the number of cells which are positive for cleaved caspase-3. Scale bar: 100 μm. *n* = 3, and 5 brain sections per mouse were analyzed and quantified. Data are presented as mean ± SEM. Data were analyzed using two-way ANOVA with Tukey’s correction. ** *p* < 0.01, *** *p* < 0.001.

**Figure 2 ijms-24-17055-f002:**
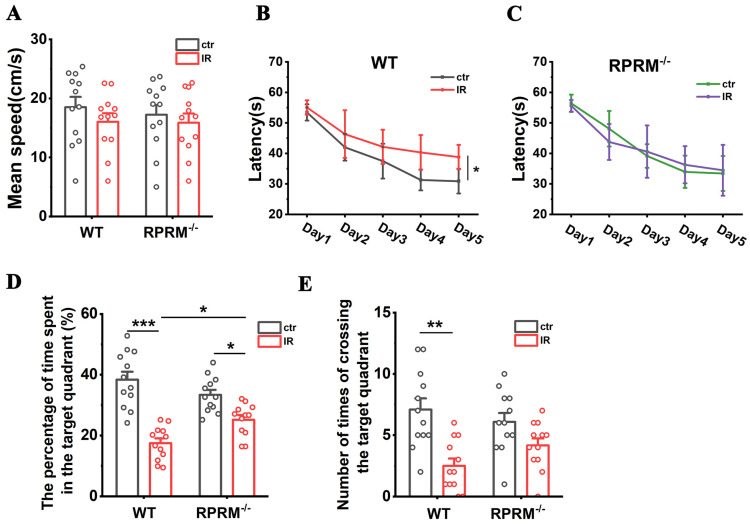
RPRM knockout alleviates radiation-induced cognitive impairment. (**A**) The average swimming speed of WT and RPRM KO mice in the place navigation test of MWM 50–54 days after WBI. (**B**) Latency of sham-irradiated and irradiated WT mice during the place navigation test of MWM. (**C**) Latency of sham-irradiated and irradiated RPRM KO mice during the place navigation test of MWM. (**D**) Percentage of exploration time of 4 groups of mice spent in the target quadrant in the spatial probe test of MWM. (**E**) Number of times 4 groups of mice crossed the target quadrant in the spatial probe test. *n* = 12. Data were presented as mean ± SEM. Data were analyzed using two-way ANOVA with Tukey’s correction. * *p* < 0.05, ** *p* < 0.01 and *** *p* < 0.001.

**Figure 3 ijms-24-17055-f003:**
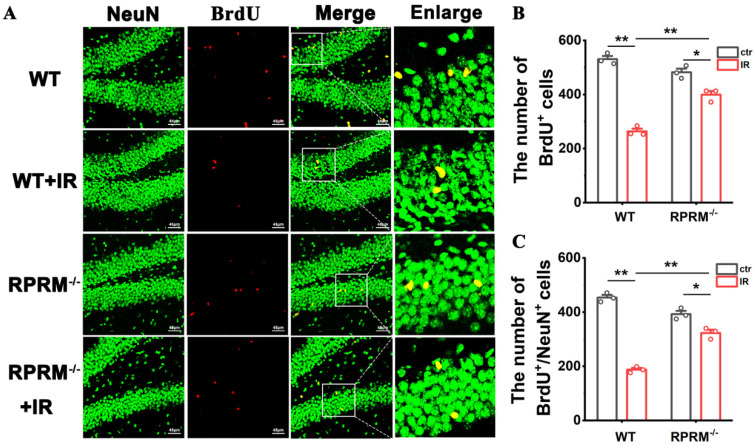
RPRM knockout alleviates radiation-induced neurogenesis inhibition. (**A**) Immunofluorescence images of BrdU+ (red) newborn cells and BrdU+ (red)/NeuN+ (green) newborn neurons in the SGZ and GCL of the DG of 4 groups of mice at 2 months post-WBI. Scale bar: 45 μm. (**B**) Quantification of the number of BrdU+ newborn cells. (**C**) Quantification of the number of BrdU+/NeuN+ newborn neurons. *n* = 3, and 5 brain sections per mouse were analyzed and quantified. Data are presented as mean ± SEM. Data were analyzed using two-way ANOVA with Tukey’s correction. * *p* < 0.05 and ** *p* < 0.01.

**Figure 4 ijms-24-17055-f004:**
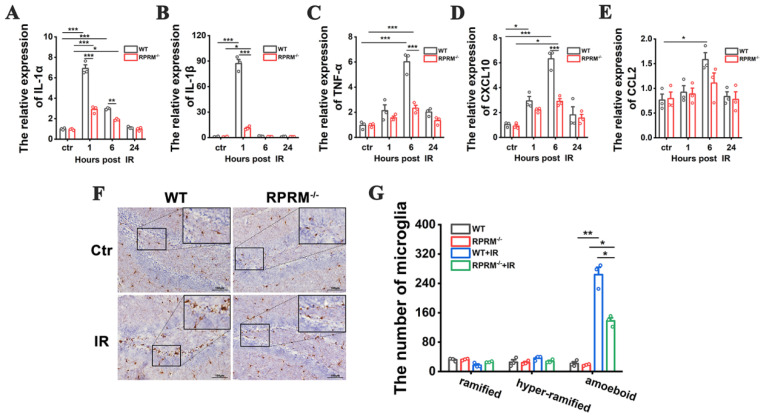
RPRM knockout diminishes radiation-induced acute neuroinflammation. (**A**–**E**) Change in the expression of proinflammatory genes of WT and RPRM KO mice within 24 h after exposure to 10 Gy WBI. *n* = 3. (**F**,**G**) Immunohistochemistry images of Iba-1 in the hippocampi of WT and RPRM KO mice at 6 h after exposure to 10 Gy WBI and quantification of the number of microglia based on their morphology. Scale bar: 100 μm. *n* = 3, and 5 brain sections per mouse were analyzed and quantified. Data are presented as mean ± SEM. Data were analyzed using two-way ANOVA with Tukey’s correction. * *p* < 0.05, ** *p* < 0.01 and *** *p* < 0.001.

**Figure 5 ijms-24-17055-f005:**
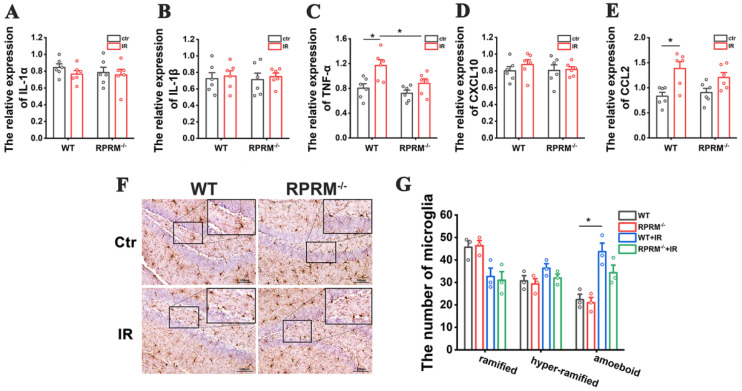
RPRM knockout attenuates radiation-induced chronic neuroinflammation. (**A**–**E**) Expression of proinflammatory genes of WT and RPRM KO mice 2 m after exposure to 10 Gy WBI. *n* = 6. (**F**,**G**) Immunohistochemistry images of Iba-1 in the hippocampi of WT and RPRM KO mice at 2 m after exposure to 10 Gy WBI and quantification of the number of microglia based on their morphology. Scale bar: 100 μm. *n* = 3, and 5 brain sections per mouse were analyzed and quantified. Data are presented as mean ± SEM. Data were analyzed using two-way ANOVA with Tukey’s correction. * *p* < 0.05.

**Figure 6 ijms-24-17055-f006:**
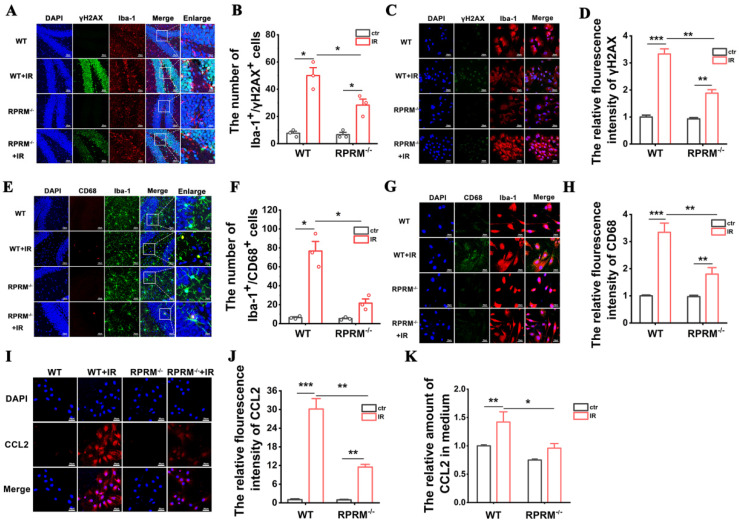
RPRM knockout attenuates the direct microglial activation induced by IR. (**A**,**B**) Images of dual immunofluorescence staining of Iba-1 (red) and γH2AX (green) in the hippocampi of WT and RPRM KO mice at 6 h after exposure to 10 Gy WBI and quantification of the number of Iba-1+/γH2AX+ cells. Blue color represents DAPI staining. Scale bar: 45 μm. (**C**,**D**) Images of dual immunofluorescence staining of Iba-1 (red) and γH2AX (green) in the primary microglia derived from WT and RPRM KO neonatal mice 6 h after 10 Gy X-irradiation and quantification of γH2AX fluorescence intensity. DAPI: blue. Scale bar: 30 μm. (**E**,**F**) Images of dual immunofluorescence staining of Iba-1 (green) and CD68 (red) in the hippocampi of WT and RPRM KO mice at 6 h after exposure to 10 Gy WBI and quantification of the number of Iba-1+/CD68+ cells. DAPI: blue. Scale bar: 45 μm. (**G**,**H**) Images of dual immunofluorescence staining of Iba-1 (red) and CD68 (green) in the primary microglia derived from WT and RPRM KO neonatal mice 6 h after 10 Gy X-irradiation and quantification of CD68 fluorescence intensity. DAPI: blue. Scale bar: 30 μm. (**I**,**J**) Images of immunofluorescence staining of CCL2 (red) in WT and RPRM KO primary microglia 6 h after 10 Gy X-irradiation and quantification of its fluorescence intensity. DAPI: blue. Scale bar: 30 μm. (**K**) The relative amount of CCL2 in the culture medium for WT and RPRM KO primary microglia 6 h after 10 Gy X-irradiation determined via ELISA. *n* = 3. Five brain sections per mouse were analyzed and quantified for (**A**,**B**,**E**,**F**). Data are presented as mean ± SEM. Data were analyzed using two-way ANOVA with Tukey’s correction. * *p* < 0.05, ** *p* < 0.01 and *** *p* < 0.001.

**Figure 7 ijms-24-17055-f007:**
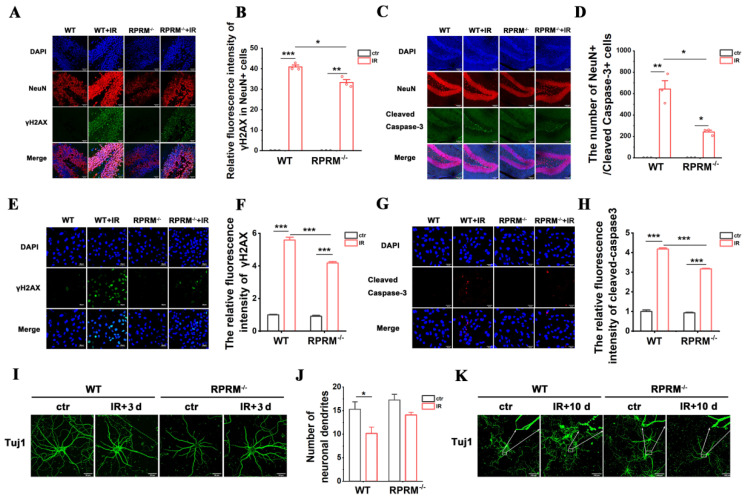
RPRM deletion protects neurons against irradiation-induced damage. (**A**,**B**) Immunofluorescence images of NeuN (red) and γH2AX staining (green) in the mouse hippocampus 6 h after exposure to 10 Gy WBI and quantification of the immunofluorescence intensity of γH2AX in NeuN+ cells. Blue color represents DAPI staining. Scale bar: 30 μm. (**C**,**D**) Immunofluorescence images of NeuN (red) and cleaved caspase-3 staining (green) in the mouse hippocampus 6 h after exposure to 10 Gy WBI and quantification of the number of NeuN+/cleaved caspase-3+ cells in the DG. Blue color represents DAPI staining. Scale bar: 100 μm. (**E**,**F**) Immunofluorescence images of γH2AX (green) in primary neurons derived from WT and RPRM fetal mice 6 h after exposure to 50 Gy X-rays and quantification of its immunofluorescence intensity. DAPI: blue. Scale bar: 30 μm. (**G**,**H**) Immunofluorescence images of cleaved caspase-3 (red) in primary neurons 6 h after exposure to 50 Gy X-rays and quantification of its immunofluorescence intensity. DAPI: blue. Scale bar: 30 μm. (**I**,**J**) Representative immunofluorescence images of Tuj1 staining in primary neurons 3 d after IR and quantification of the number of neuronal dendrites. Scale bar: 30 μm. (**K**) Representative immunofluorescence images of Tuj1 staining in primary neurons 10 d after IR. Scale bar: 100 μm. Data were analyzed using two-way ANOVA with Tukey’s correction. * *p* < 0.05, ** *p* < 0.01, *** *p* < 0.001 compared with the relative control. *n* = 3.

**Figure 8 ijms-24-17055-f008:**
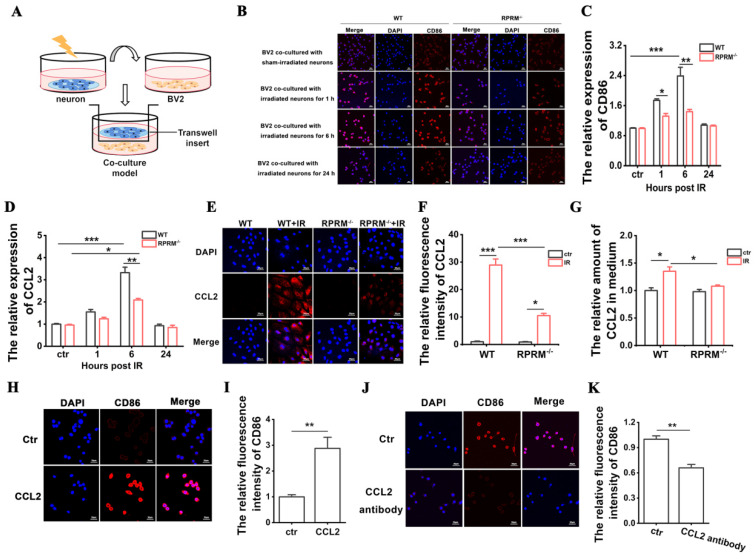
RPRM knockout attenuates the microglial activation induced by irradiated neurons. (**A**) Schematic diagram of co-culture system for BV2 cells and primary neurons. (**B**,**C**) Representative immunofluorescence images of CD86 (red) in BV2 cells after co-culture with irradiated primary RPRM^+/+^ and RPRM^−/−^ neurons for different times and quantifications of its immunofluorescence intensity. Blue color represents DAPI staining. Scale bar: 30 μm. (**D**) Change in the mRNA expression of CCL2 in the primary RPRM^+/+^ and RPRM^−/−^ neurons after 50 Gy X-irradiation. (**E**,**F**) Representative immunofluorescence images of CCL2 (red) in the primary RPRM^+/+^ and RPRM^−/−^ neurons 6 h after 50 Gy X-irradiation and quantification of its immunofluorescence intensity. DAPI: blue. Scale bar: 30 μm. (**G**) The relative amount of CCL2 in the culture medium for the primary RPRM^+/+^ and RPRM^−/−^ neurons determined 6 h after 50 Gy X-irradiation. (**H**,**I**) Representative immunofluorescence images of CD86 (red) in BV2 cells treated with recombinant CCL2 protein and quantification of its immunofluorescence intensity. DAPI: blue. Scale bar: 30 μm. (**J**,**K**) Representative immunofluorescence images of CD86 (red) in BV2 cells co-cultured with irradiated WT neurons for 6 h in the absence and presence of CCL2 antibody and quantification of its immunofluorescence intensity. DAPI: blue. Scale bar: 30 μm. *n* = 3. Data are presented as mean ± SEM. Data were analyzed using two-way ANOVA with Tukey’s correction for (**B**–**G**) and paired *t*-test for H–K. * *p* < 0.05, ** *p* < 0.01 and *** *p* < 0.001.

**Table 1 ijms-24-17055-t001:** List of primers used for qPCR.

GAPDH	forward primer	TGACCACAGTCCATGCCATC
	reverse primer	GACGGACACATTGGGGGTAG
CCL2	forward primer	AGGTGTCCCAAAGAAGCTGTAG
	reverse primer	AATGTATGTCTGGACCCATTCC
IL-1α	forward primer	AGATGGTCAATGGCAGAACTGT
	reverse primer	CGCTTGAGTCGGCAAAGAAA
IL-1β	forward primer	TGCCACCTTTTGACAGTGATG
	reverse primer	TGTGCTGCTGCGAGATTTGA
TNF-α	forward primer	CAAATTCGAGTGACAAGCCTG
	reverse primer	GAGATCCATGCCGTTGGC
CXCL10	forward primer	CAAGCCATGGTCCTGAGACA
	reverse primer	TGAGCTAGGGAGGACAAGGA

## Data Availability

All experimental data reported in this article will be shared by the corresponding author on request.

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
