# Peer review of "Reprimo (RPRM) as a Potential Preventive and Therapeutic Target for Radiation-Induced Brain Injury via Multiple Mechanisms"

_ijms, 2023, doi:10.3390/ijms242317055_

Round 1
Reviewer 1 Report
Comments and Suggestions for Authors
The manuscript submitted by Zhujing Ye and coauthors is devoted to investigating the effect of Reprimo gene knockout on the functioning of brain cells in in vitro and in vivo models of radiation-induced brain injury. The study is complex and performed at a high methodological level. Undoubtedly, it can be recommended for publication after revisions.
- The authors describe in the materials and methods section that the primary cultures contain "neuronal" cells. What do the authors mean using this term: only neurons or neurons and glial cells? In the second case, some authors use the more appropriate term "neural". Notably, all the primary cultures, including those grown in Neurobasal+B27 medium, contain astrocytes, but the term “astrocytes” is used only in the Introduction. Please clarify this misleading point.
- Regarding the previous comment, please clearly indicate the molar concentration of glutamine in the culture medium (line 98).
- The authors used only 3 mice per group in some experiments. So small sample size in experiments with animals may raise questions about the data reliability. Please describe the sample size estimation method (software, parameters, etc.) used in the present study.
- Thoroughly check the text and correct typos, including in the abstract (Taken together, Our results).
Comments on the Quality of English LanguageOnly minor editing is required.
Reviewer 2 Report
Comments and Suggestions for Authors
The authors have submitted a research article of demonstrating an impact of knockout of the gene for Reprimo (RPRM), a tumor suppressor gene, on levels of radiation-induced brain injury in animal models lacking RPRM. It has been well recognized that exposure of cranium with therapeutic radiation against brain tumors could increase the risk of radiation-induced brain injury in patients with brain tumors. In this regard, their study process found statistically positive associations of the loss-of-function of RPRM with decrease in the levels of radiation-induced brain injury, which might have a potential to be an important knowledge regarding suppression of brain injury after radiation by the loss-of-function of RPRM. This issue is of interest, and impact of their article is strong. My overall concern with the article describing the available data regarding a possible, beneficial availability of RPRM which might predict risk of radiation-induced brain injury in patients with brain tumors is that information provided may offer something substantial that helps advance our understanding of effective management of radiation-induced brain injury which draws novel class of effective medicinal compounds available in clinic.
The therapeutic prediction based on the present data is definitely the primary goal of this submission, so that readers are likely to know how to predict the effective compounds on RPRM function, based on the information regarding the data like this study. The authors should discuss this issue in their revision.
In addition to that issue, to strengthen authors’ perspectives, the authors are strongly recommended to add a “toxicology” discussion in detail regarding known loss-of-function of RPRM effect on humans (or even on rodents), for instance. The opposite, toxicological effects of expected outcomes, if known, may influence largely the authors’ perspective.
Reviewer 3 Report
Comments and Suggestions for Authors
Thank you for the opportunity to review this manuscript. The manuscript is very well written and easy to understand, addresses an area of critical need with very limited treatment options.
The experimental design is clear and the presentation is very logical. I appreciated how the investigators tested both acute and delayed radiation related injury. I also appreciated use of sham irradiation as a control for both wildtype and RPRM knockout groups.
The experiments consider multiple mechanisms of injury with consideration of acute and long term effects, impact on inflammatory markers, and microglia.
Some thoughts for consideration:
1- In addition to the graphical representation of the numerical results in various results figures, if possible, it would be welcome for the manuscript to provide median values and confidence intervals within the text or as a supplement, with indication of where statistically significant differences were observed.
2- In figure 2A, the difference between swimming speeds appears minimal between the two groups. Please indicate in the results how many days after radiation this was tested.
3- The latency testing in figure 2 is carried out to day 5. Later in figure 5, data is carried out to 2 months after radiation exposure. It would have been interesting to see effects on cognitive performance testing more than 5 days after end of RT.
4- In the discussion, further comments related to how the evaluated mechanism can be utilized as a therapeutic target would be welcome.
5- While prevention of radiation induced injury is important, it is also vitally important to achieve this without compromising antitumor efficacy of radiation therapy. Do the findings suggest there might be an issue in this regard with use of RPRM as a possible therapeutic target?
Overall, I think this is an excellent manuscript and these findings should be published.
Round 2
Reviewer 1 Report
Comments and Suggestions for Authors
All my comments have been addressed.
Reviewer 2 Report
Comments and Suggestions for Authors
The authors have done a good job responding to reviewer comments and concerns in their revision. I believe the manuscript is significantly improved as a result. Now I recommend that this revised version of the manuscript can be accepted for publication in IJMS.